# South Korean Nurses’ Experiences of Speaking up for Patient Safety and Incident Prevention

**DOI:** 10.3390/healthcare11121764

**Published:** 2023-06-15

**Authors:** Jeong Hee Jeong, Sam Sook Kim

**Affiliations:** 1Department of Nursing Science, Kyungsung University, Busan 48434, Republic of Korea; loveu1105@ks.ac.kr; 2Department of Nursing, Daedong College, Busan 46270, Republic of Korea

**Keywords:** incidents, nurse, patient safety, qualitative study, speaking up

## Abstract

Despite the importance of speaking up for patient safety, hesitancy to do so remains a major contributing factor to communication failure. This study aimed to investigate the experiences of South Korean nurses in speaking up to prevent patient safety incidents. Twelve nurses responsible for patient safety tasks or with experience in patient safety education were recruited from five hospitals (three university hospitals, two general hospitals) in city “B”. Data were collected through open-ended questions and in-depth interviews, transcribed, and analyzed using the inductive content analysis method. The study resulted in the identification of four main categories and nine subcategories that captured commonalities among the experience of the 12 nurses. The four main categories were as follows: the current scenario of speaking up, barriers to speaking up, strategies for speaking, and confidence training. There is a scarcity of research on speaking-up experiences for patient safety among nurses in South Korean. Overall, it is necessary to overcome cultural barriers and establish an environment that encourages speaking up. In addition, developing speaking-up training programs for nursing students and novice nurses is imperative to prevent patient safety incidents.

## 1. Introduction

Patient safety is critical to delivering high-quality and essential healthcare services [1]. It focuses on reducing the probability of preventable adverse events and medical errors that occur during hospital care. Effective communication is the key to managing such errors [2,3]. “Speaking up” refers to the expressing of concerns or disagreement for the benefit of patient safety by a healthcare professional who recognizes or becomes aware of another healthcare professional’s incompetent or dangerous behavior, incorrect diagnosis, questionable clinical judgment, nonadherence to rules, or shortcuts that threaten patient safety [4,5]. Speaking up helps intercept errors, improves system failures, and promotes patient safety and healthcare quality [6]. In recent years, interest in speaking up about patient safety concerns has been increasing [3].

Given that nurses hold a unique position in recognizing and improving patient safety [7], it is essential for them to be willing to speak up and prevent adverse events when a patient is not receiving optimal treatment [8]. However, research indicates that nurses exhibit lower psychological stability when it comes to speaking up for patient safety and display a high level of resignation compared to other healthcare staff, including doctors [9]. They often seek quick, short-term solutions instead of investigating the underlying cause [8]. Failure to speak up can also be attributed to the fear of retaliation, reprisal, negative feedback, or appearing incompetent; prior negative outcomes; and various obstacles to effective communication [3,6,10]. Communication failure among healthcare professionals is one of the main causes of medical errors and patient harm [10]. In a Danish report analyzing the root cause of communication errors, hesitancy to speak up was seen in 23% of hospital staff [11]. Thus, understanding the factors influencing speaking up is essential to improve communication.

Speaking up provides a great opportunity to proactively prevent incidents. Moreover, it allows medical professionals to analyze the factors that motivate or discourage speaking up and develop strategies to improve speaking-up behaviors. These factors include having the confidence to speak up [12], the perceived safety of speaking up, and the overall attitude toward patient safety and teamwork [13,14]. Intervention studies [12,13,14,15,16,17] and systematic reviews of articles on speaking up [6,18] have been conducted in various countries. However, studies on speaking-up experiences among nurses in South Korea are limited. Speaking up is particularly challenging for South Korean nurses because of cultural and social factors, including hierarchy, etiquette, and manners [18,19].

This study aimed to examine experiences among nurses in South Korea of speaking up to prevent patient harm and to provide baseline data for developing training materials for effectively speaking up for patient safety, especially for novice nurses and nursing students who are more likely to remain silent.

## 2. Materials and Methods

### 2.1. Research Design

This exploratory descriptive qualitative study used open-ended questions and one-on-one in-depth interviews to explore and describe speaking-up experiences among participants.

### 2.2. Participants

This study included the experiences of 12 nurses working at three university hospitals and two general hospitals in city “B”, South Korea. The inclusion criteria were as follows: (1) nurses overseeing patient safety tasks or with experience in patient safety education, (2) nurses recommended by the nursing department, and (3) those who agreed to participate in the study. Nurses with extensive patient safety experience were specifically selected as participants due to their long work experience and high positions, enabling them to provide comprehensive and objective insights into nurses’ experiences of speaking up within the context of overall patient safety management. Purposive sampling was employed to recruit participants. The researchers visited the nursing department of five hospitals to explain the study plan and selection criteria and to obtain participant recommendations. The purpose, procedure, and the rights of the research participants were fully explained in both written and oral forms to the nurses recommended by the nursing department, and voluntary consent was obtained.

### 2.3. Data Collection

Data collection took place between January and April 2022. Building upon previous studies on speaking up [16,17,18] and the researchers’ expertise in safety incident prevention, interview questions were designed to elicit comprehensive and in-depth accounts of nurses’ experiences in speaking up to prevent safety incidents. The questionnaire was revised with the help of qualitative research experts. The questions comprised four types: introductory, transitional, main, and closing. The main questions were as follows: (1) “Have you heard of speaking up?”, (2) “Have you spoken up in a situation when patient safety was in danger?”, and (3) “What systems need to be in place to encourage speaking up?” In the closing section, the discussed content was summarized, and the participants were provided with an opportunity to make additional comments or ask questions. More than one individual phone interview was conducted. After checking and analyzing the collected data, data collection was discontinued once it was judged that no new information was identified and that the data were sufficiently saturated.

### 2.4. Data Analysis

The transcribed data were analyzed using the inductive content analysis process as described in a previous study [20]. In the first phase, the transcribed data were read repeatedly to derive units of analysis by selecting meaningful characters, words, sentences, and paragraphs by comprehending the overall context of the research questions and collected responses. In the second phase, the process was refined by repeatedly reading the selected units of analysis to create open coding transcripts using concepts or phrases for appropriate headings. In the third phase, subcategories were derived by comparing the codes and categories to determine the similarities and differences between the created codes. In the last phase, an abstraction process was performed to categorize the codes into an inclusive category by re-examining the relationship between categories and the different types and attributes of the main narratives.

### 2.5. Questionnaire Validity and Quality Assurance

This study used the qualitative research checklist developed by Nacioglu [21] to ensure quality. In the preparation phase, the data collection and sampling strategies were examined. In the organization phase, the categorization, abstraction, interpretation, and representativeness of the data were checked. In the reporting phase, whether the process of analyzing and reporting was systematic and logical was examined, as well as how the data related to the results. Throughout all phases, two experts in qualitative research (one professor of nursing and one professor of pedagogy) were consulted. In addition, to ensure the transferability of the research results, feedback from two nurses outside of the study who had similar experiences as the study participants was requested. After reading the study findings, both external nurses agreed with the study participants and provided positive feedback. Furthermore, to establish the validity and reliability of this study, the qualitative researcher strived to ensure trustworthiness by reviewing various studies and checking each phase in detail.

### 2.6. Ethical Considerations

This study was approved by the Institutional Review Board (IRB) of Kyungsung University (IRB No. KSU-21-12-003) and conducted in accordance with the principles of the Declaration of Helsinki. All participants were notified that the conversations were recorded. To guarantee the anonymity of the participants while differentiating them, each participant was assigned a number. All participants were provided a small gift for their participation.

## 3. Results

### 3.1. Participant Characteristics

A total of 12 nurses (eight from university hospitals and four from general hospitals) participated in this study. All participants were female, and their average experience in patient safety was 9.3 years. The medical institutions where the participants work primarily cater to patients with severe diseases and have established patient safety management organizations. The organizational culture and nursing system in these medical institutions are characterized by strict hierarchies (Table 1).

### 3.2. Speaking-Up Experiences

Four categories and nine subcategories of commonalities among the speaking-up experiences of the nurses were derived (Table 2).

#### 3.2.1. Current Scenario of Speaking Up

Hospitals are implementing speak-up campaigns aimed at patients and caregivers to prevent patient safety incidents. However, the focus primarily remains on the patient safety incident reporting system for healthcare workers.

All participants emphasized the importance of speaking up for patient safety. Some participants mentioned that the current speak-up campaign primarily promotes patient double-checking procedures. This includes asking patients and their guardians questions related to their treatment to ensure their safety and the safety of their families, thus preventing patient safety incidents.

*“I’ve never heard of ‘speak up’. For patient safety, patients and guardians are encouraged to speak up through the speak-up campaign. However, there is no speak-up campaign targeting medical personnel.”* (P1)

*“Patients and guardians are being taught to ask questions to medical staff when they have questions regarding patient identification, prescription, treatment, and surgery, and to speak up when checking patients.”* (P3)

Most participants said that there is no speak-up system for healthcare workers. However, they stated that patient safety management is being carried out, such as double checking by medical personnel before blood transfusion, the participation of surgical team members in a time-out before surgery, and patient checks before medication administration.

*“There is no specific system or policy for speaking up among medical staff. Just for patient safety, we are forced to stop working to check patient information through time-out (participation of surgical team members) before blood transfusion and before surgery.”* (P6)

*“Speak up for medical professionals is currently in the stage of publicizing and promoting it. However, through patient confirmation procedures, confirmation before blood transfusion (two or more medical personnel), timeout during surgery (participation of surgical team members), and matching of patient information before administration are checked.”* (P7)

#### 3.2.2. Barriers to Speaking Up

Participants said that the barriers to speaking up were fears that personal relationships would be damaged by communicating their concerns or a tendency to remain silent due to a hierarchical organizational culture. Participants said that nurses tended to blame themselves or ignore when they witnessed an act that threatened patient safety. In addition, some participants understood that it would be difficult for nurses to speak up publicly because speaking up to others was perceived as excessive interference or pointing out mistakes.

*“Nurses will be angry with themselves, blame themselves, and regret from the moment they recognize that it is a mistake if they have performed a nursing act that threatens patient safety. However, it would be difficult to speak up about the wrongdoings of others as it would be like meddling.”* (P2)

*“Speaking up is difficult because it seems like pointing out the other person’s mistakes or faults.”* (P10)

*“Some nurses just ignore it or ignore it if it is not directly related to them.”* (P9)

Participants said that they had experience speaking up in situations when they held a higher position or higher level of experience than their counterparts or belonged to the same affiliation. However, it was recognized that it was burdensome to raise objections in situations related to doctors, senior nurses, and other occupations.

*“I am also a head nurse now, so I can speak up to the doctor and ward nurses, for example, when a sterile technique is omitted. When I was a new nurse, I couldn’t say anything.”* (P3)

*“It is not easy for junior nurses to speak up about the senior nurse’s misbehavior. In most cases, senior nurses can speak to junior nurses in a scolding tone about any wrongdoing.”* (P6)

*“There have been cases where I (the head nurse) spoke up to the nurses on the ward, but there was no case where I presented a counterargument to the unfair behavior of other healthcare workers. I think it is difficult to speak up at the moment because the atmosphere that accepts speaking up systematically (between health care workers) has not been formed.”* (P7)

#### 3.2.3. Strategies for Speaking Up

Participants emphasized the need for a speak-up system that focuses on patient safety, advocating for publicity, systematic education, and dissemination efforts. They unanimously recognized speaking up as a necessary system to prevent patient safety incidents. Participants suggested that changing the perception of patients, staff, and medical personnel is of utmost importance in establishing a speak-up culture. To achieve this, they proposed plans for ongoing publicity activities, including announcements and information sharing related to speaking up.

*“First of all, the perception of speaking up among patients, staff, and medical staff must change. In particular, employees do not know the speaking up. Therefore, patient safety organizations or staff should promote the benefits of speaking up. And continuous promotional activities are also necessary.”* (P12)

*“I think that speaking up is a change in awareness and participation that is necessary for patient safety. Information and announcements about speaking up should attract the attention and activity of all employees. For example, it is suggested that improvements to patient safety incidents be communicated and shared with staff through a speaking up best practice presentation”* (P8)

Some participants mentioned that since speaking up is a sensitive matter that can cause misunderstanding, empathy from and education for all employees are needed. Some participants included speak-up education as a patient safety activity and suggested specific education methods, procedures, and content.

*“First of all, it seems that education and understanding of speaking up are needed for all employees. If speaking up is wrong, it can cause conflict between occupations and classes, so it would be nice to have a situation, stage, and specific format in which to speak.”* (P5)

*“Speak-up training should be mandatory for all employees to participate and speak up. If speak-up education is established as a patient safety system, I hope it becomes an essential procedure that all employees must perform. For example, I want it to be like Time Out before surgery and Sign Out after surgery.”* (P6)

Participants expressed their hope that speaking up would serve as a preventive measure to mitigate patient safety incidents. Some participants highlighted the cost and time-intensive nature of the patient safety reporting system, suggesting that speaking up could facilitate a patient safety culture where immediate solutions can be implemented.

*“The hospital has implemented a patient safety reporting system aimed at preventing and improve patient safety. Nevertheless, hundreds of cases, such as proximity errors and adverse events, occur every year. We need to come up with a more specific and detailed method. In particular, speaking up means to present opinions on threatening factors before a patient safety incident occurs, and if a culture that can accept this is spread, I think it is a way to reduce patient safety incidents.”* (P1)

*“The current patient safety reporting system requires a lot of economic cost and time to improve. On the other hand, speaking up has the advantage of being able to improve behavior immediately and rapidly without any restrictions on cost, place, or time. So, I hope that a culture that accepts speaking up will spread.”* (P10)

*“I think that speaking up must be established to facilitate a culture that prioritizes patient safety. In hospitals, various occupations are related to patient safety, so all organizations must work together. If open and proactive speak-up behavior settles into a patient safety culture, I think speaking up will be accepted as a natural procedure.”* (P11)

### 3.3. Confidence Training

All participants recognized that nursing students require speak-up education for patient safety. In addition, various plans for speak-up education methods, content, and timing were presented. All participants said that speak-up education for nursing students was necessary. In particular, speak-up education was proposed to enhance communication skills when student nurses face a situation that requires them to speak up during hospital practice.

*“In some cases, student nurses witness a situation that threatens patient safety at the hospital practice site, or they themselves are involved. We need speak-up education to deal with these situations.”* (P2, P8)

*“Speaking up is often done in unfair and wrong situations, so communication skills are needed to express one’s thoughts and opinions without damaging relationships with others. For example, you need skills to communicate proficiently with staff, patients, and caregivers.”* (P8, P1)

Participants suggested incorporating role-playing methods into speak-up education for nursing students, specifically focusing on medication errors, blood transfusion incidents, surgeries, and procedures associated with high patient safety risks.

*“It is recommended to virtually set up a speak-up situation (e.g., medication error, blood transfusion incident, surgery, procedure, fall, etc.) and experience the role of a patient, nurse, or doctor. I think a kind of role play could be an alternative.”* (P7)

*“When designing a speak-up education case, I think it can only be effective if it includes difficult ward patients and staff, unexpected questions, and the consequences of speaking up success and failure.”* (P4)

## 4. Discussion

This study aimed to explore the experiences and perceptions of speaking up for patient safety among South Korean nurses. The findings revealed that Korean nurses primarily rely on the official patient safety reporting system rather than engaging in direct speaking up when faced with situations that require their voice. Our discussion centers around four commonalities observed in the speak-up experiences highlighted in this study: the current scenario of speaking up, barriers to speaking up, strategies for speaking up, and confidence training.

In healthcare facilities, the focus of speaking up for patient safety has primarily been on patients and caregivers, with an emphasis on teaching and training them to actively participate in their medical care by asking questions and confirming the care they receive. However, healthcare professionals themselves often hesitate to speak up in situations that go beyond the guidelines set by the Korea Institute for Healthcare Accreditation, which aim to prevent safety incidents (e.g., double-checking patient information before surgery, medication administration, or transfusion). This hesitation may stem from the burdensome process of reporting patient safety incidents. It is crucial to recognize that speaking up among healthcare professionals is essential for preventing medical errors and ensuring patient safety, as hesitancy in speaking up can jeopardize patient well-being and lead to incidents [6]. As such, a medical facility needs proactive leadership and policy-based managerial support that encourages voicing concerns about the hospital environment among healthcare professionals [22]. They also need to provide a safe environment for speaking up freely and comfortably [18]. In addition, it is necessary for the Korea Institute for Healthcare Accreditation to include speaking up as one of the accreditation criteria so that patient safety competency among healthcare professionals can be improved and adverse events prevented.

Traditionally, medical institutions in South Korea are hierarchical organizations. As such, there is a tendency to remain silent or avoid speaking up to a more senior member of staff, even in situations endangering patient safety. The barriers to effective communication about patient safety concerns include fear of meddling with someone else’s job responsibilities, fear of non-acceptance or changes among doctors and other high-level healthcare providers, and fear of a hostile work environment after speaking up [23]. Organizations with a culture of encouraging members to voice patient safety concerns continue to promote speaking-up behaviors. In contrast, remaining silent discourages speaking-up activities [24]. Therefore, it is important that medical institutions support speaking-up training for nurses to ensure stable organizational performance by reinforcing their levels of assertiveness and speaking-up behaviors [25].

In addition, open and supportive leadership, a psychologically safe work environment based on a trusting relationship between a nurse manager and floor nurses [18,24], and assertiveness training [26] were found to increase willingness to speak up. Therefore, it is necessary to implement leadership training for managers and assertiveness training for novice nurses. As such, to transform the prominent hierarchical culture within the medical environment, the development and implementation of multiphase and multifaceted educational programs that involve all organizational members are vital. Such programs are expected to build confidence among individuals to speak up about patient safety and lead to continuous interaction among medical professionals and a cycle of positive feedback, ultimately improving the quality of patient safety and establishing a culture for open communication.

The study participants reported the occurrence of ongoing patient safety incidents, despite the presence of patient safety management systems in their medical facilities. They expressed a strong desire to contribute to a work culture that prioritizes patient safety through speaking up. Hospitals adhere to the national protocol for incident reporting, analysis, and implementing corrective measures, and they also share reports on patient safety incidents. However, statistics revealed that, in 2021 alone; South Korea experienced 13,146 patient safety incidents [27]. While it may be challenging to determine whether this number accurately represents the full extent of patient safety incidents, it undeniably underscores the ongoing need for continuous efforts to enhance the quality of patient safety management. Moreover, the study participants emphasized the importance of educating healthcare professionals about the overall concept of speaking up and providing them with guidance on effective communication strategies.

The effect of speak-up training has been previously validated [16,17]. In one study, participating nurses were shown five real-world clinical cases of failure to speak up and the resulting negative impact on patient safety. After the intervention, there was a change in perception and behavior regarding speaking up among participants, with most resolving to refuse a physician’s unsafe order [16]. In another study, simulation-based training for novice anesthesiology trainees significantly improved their scores in self-efficacy, social outcome expectations, and self-assertive attitude [17]. As such, implementing or applying a proper educational intervention program can induce a positive change in healthcare professionals’ perceptions of speaking up. Therefore, it is imperative to develop situational scenarios by considering the expertise and attributes of healthcare professionals and establishing a training program tailored to them.

The study participants emphasized that training nursing students on how to speak up will help improve the quality of health care. As a way of cultivating knowledge, skills, and attitudes toward speaking up, they also suggested using clinical cases and building communication skills to help nursing students voice their ideas and opinions in circumstances that require them to speak up. Although nursing students witness suboptimal practice in clinical settings, they often remain silent because of a lack of moral courage, a fear of retaliation, a desire to maintain a good relationship with their mentors, and their low status as students. As such, speaking up is an assertive communication skill required in clinical situations and is intricately connected with personal, organizational, social, and cultural factors [28].

Healthcare facilities and universities need to provide training to nursing students on patient safety and speaking up assertively to improve their confidence and ability to voice their concerns [26]. Previously, simulation-based training for speaking up was found to be effective for nursing students [13,14]. Thus, training should be based on simulation programs to reflect diverse and complex clinical situations, while ensuring a safe practice environment.

This study is subject to several limitations. First, the data collection was limited to hospital nurses from a single region, which may restrict the generalizability of the findings to nurses in other medical institutions and regions. Therefore, caution should be exercised when interpreting the results of this study. Second, the study did not specifically assess the effectiveness and success of speaking up in preventing patient safety incidents. In future research, a comprehensive evaluation of the speaking-up situations faced by nurses is warranted. Additionally, further studies investigating nurses’ experiences in speaking up for the prevention of patient incidents are needed.

## 5. Conclusions

This study identified four categories pertaining to the perceived experiences of speaking up for the prevention of patient incidents among South Korean nurses: (1) the current scenario of speaking up, (2) barriers to speaking up, (3) strategies for speaking up, and (4) confidence training. These findings have important implications for identifying pertinent issues related to speaking up among nurses and serve as a foundation for developing strategies to enhance speak-up behavior.

The findings indicate the need to overcome cultural and societal barriers and establish an environment in which speaking up is encouraged. In addition, it is imperative to develop speak-up training programs for nursing students and novice nurses to prevent patient incidents. Moreover, studies that explore the inclusion of healthcare professional speak-up behavior as a criterion of the Korea Institute for Healthcare Accreditation are warranted.

## Figures and Tables

**Table 1 healthcare-11-01764-t001:** Participant characteristics.

Participant Number	Current Workstation	Position	Patient Safety Experience	Type of Medical Institution	Level of Patient Safety Certificate
1	Quality improvement	QI director	12 years	University hospital	Accredited by medical institution
2	Internal medicine ICU	Senior nurse	10 years	University hospital	Accredited by medical institution
3	Trauma center	UM	10 years	University hospital	Accredited by medical institution
4	Quality improvement	Patient safety specialist	10 years	University hospital	Accredited by medical institution
5	Respiratory ICU	UM	15 years	University hospital	Accredited by medical institution
6	Quality improvement	Patient safety specialist	8 years	University hospital	Accredited by medical institution
7	Research support team	Team lead	10 years	University hospital	Accredited by medical institution
8	Nursing department	Nurse trainer	12 years	University hospital	Accredited by medical institution
9	Quality improvement	Team lead	8 years	General hospital	Accredited by medical institution
10	Quality improvement	Team lead	8 years	General hospital	Accredited by medical institution
11	Quality improvement	Patient safety specialist	4 years	General hospital	Accredited by medical institution
12	Quality improvement	Patient safety specialist	4 years	General hospital	Accredited by medical institution

Abbreviations: ICU, intensive care unit; UM, unit manager.

**Table 2 healthcare-11-01764-t002:** Experience of speaking up.

Category	Subcategory
Current scenario of speaking up	Speaking up centered on patients and guardians
	Healthcare worker-centered patient safety management
Barriers to speaking up	Personal avoidance
	Hierarchical organizational culture
Strategies for speaking up	Promoting the speaking up behavior system
	Systematic speaking up behavior education
	Expansion of the speaking up system
Confidence training	Communication-based speaking up skills
	Clinical case-based speak-up training

## Data Availability

Not applicable.

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
