# Peer review of "South Korean Nurses’ Experiences of Speaking up for Patient Safety and Incident Prevention"

_healthcare, 2023, doi:10.3390/healthcare11121764_

Round 1
Reviewer 1 Report
Overall, the paper is well written, and theoretically informed.
Please elaborate on the identification and justification of selected participants. Readers want to know the reason for female predominance in the study and why almost all the group was composed of non-young and non-minority participants. Mention the inclusion/exclusion criteria for selecting participants.
Please elaborate more on the qualitative analysis. It is important to know how you moved from codes to themes (e.g. inductively? deductively?)”
The results of the study are mostly descriptive and there is limited analysis. There is also absence of thick description, which one would expect in a qualitative study.
The data provided are insufficient to warrant being called a theme… use sufficient quotes to support theme/sub-theme.
Author Response
Thank you for your review, which has helped us improve the quality of this manuscript. According to the reviewers’ comments, we have made revisions to the manuscript. We have used red-colored fonts to indicate the revised parts.
|
Materials and Methods |
|
|
Comments |
Response |
|
1. Please elaborate on the identification and justification of selected participants. Readers want to know the reason for female predominance in the study and why almost all the group was composed of non-young and non-minority participants. Mention the inclusion/exclusion criteria for selecting participants. 2. Please elaborate more on the qualitative analysis. It is important to know how you moved from codes to themes (e.g. inductively? deductively?) |
1. The text has been amended to clarify the inclusion criteria (lines 80-87). Since most South Korean nurses are female, it was difficult to include males in the study. In addition, nurses with patient safety experience have long working experience and are in high positions. 2. The text has been amended to reflect the reviewer’s comments (lines 16, 113). Inductive content analysis was employed as the research method. |
|
Results |
|
|
1. The results of the study are mostly descriptive and there is limited analysis. There is also absence of thick description, which one would expect in a qualitative study.
2. The data provided are insufficient to warrant being called a theme… use sufficient quotes to support theme/sub- theme. |
1. 1) Participants were not familiar with the term “speak up.” 2) Participants had a combination of direct and indirect experiences. 3) An additional phone interview was attempted to elicit more vivid experiences. However, participants expressed concerns about the potential exposure of their medical institutions and staff’s lack of patient safety and accident prevention practices. Particularly, addressing patient safety accidents is a highly sensitive issue. Therefore, there were limitations in conducting a comprehensive and in-depth analysis.
2. The text has been amended to reflect the reviewer’s comments (lines 246-247, 251-253). The necessary revisions were made to the quotations of participants 12 and 8, and they have been reanalyzed and are now presented in the manuscript. |
Reviewer 2 Report
Thank you for the opportunity to review an innovative and captivating article.
The article is excellently written, addressing crucial aspects of patient safety that are often overlooked or not adequately discussed.
However, in line 43 of the introduction, it would be beneficial to provide more details beyond simply stating "MOST NURSES." Expanding on the specific countries or regions where this topic was examined and incorporating additional sources to support this claim would strengthen the article's credibility.
The findings presented in the article are precise and well-supported, providing valuable insights into the subject matter.
The discussion section of the article is comprehensive and effectively stimulating renewed discourse on the topic.
Author Response
Thank you for your review, which has helped us improve the quality of this manuscript. According to the reviewers’ comments, we have made revisions to the manuscript. We have used violet-colored fonts to indicate the revised parts. Overlapping reviewer's comments are specifically marked in red within the revised manuscript.
|
Introduction |
|
|
Comments |
Response |
|
1. In line 43 of the introduction, it would be beneficial to provide more details beyond simply stating "MOST NURSES." Expanding on the specific countries or regions where this topic was examined and incorporating additional sources to support this claim would strengthen the article's credibility. |
1. "Most nurses" was removed and nurses' speaking up was described based on the comparison results between nurses and other healthcare staff (lines 42-45). |

Reviewer 3 Report
Why only focusing in Nurses and not including other healthcare professionals? Medical errors can be done by other profesisonals...
Have you explored some sub-category for assessing patienst' involvement?
The barriers were assessed, but what about their streghts for speaking about safety?
Were there any differences in the speachs of the senior (leading) staff and the junior staff?
The qualitatve research should be interpreted in the context where it has been developed. As such, authors should provide a context of the institutions and profession culture in those settings.
Patient safety "accidents" are not such thing. It is about incidents. Please, revise
Author Response
Thank you for your review, which has helped us improve the quality of this manuscript. According to the reviewers’ comments, we have made revisions to the manuscript. We have used green-colored fonts to indicate the revised parts. If the reviewer's comments overlap, they are indicated in red.
|
Materials and Methods |
|
|
Comments |
Response |
|
1. Why only focusing in Nurses and not including other healthcare professionals? Medical errors can be done by other profesisonals. |
1. This study focused on exploring nurses' speaking up experience. Comparisons with other healthcare staff were not the focus of this study. |
|
Results |
|
|
Comments |
Response |
|
1. Have you explored some sub-category for assessing patienst' involvement?
2. The barriers were assessed, but what about their streghts for speaking about safety?
3. Were there any differences in the speachs of the senior (leading) staff and the junior staff?
4. The qualitatve research should be interpreted in the context where it has been developed. As such, authors should provide a context of the institutions and profession culture in those settings. |
1. Participants consisted of nurses. There was no patient involvement. I trust this addresses your concern. If not, kindly clarify and I will provide a suitable response. 2. Some participants mentioned a sense of responsibility and courage for patient safety. However, their strengths could not be derived due to a lack of sufficient citations. 3. The text has been amended to reflect the reviewer’s comments (line 228). The entire quotation of participant number six was reanalyzed, improved, and it has been reflected. 4. The text has been amended to reflect the reviewer’s comments (lines 155-159). Additional comments were made on the environment and organizational culture of medical institutions. |
|
The Quality of English Language |
|
|
Comments |
Response |
|
1. Patient safety "accidents" are not such thing. It is about incidents. Please, revise. |
1. In the manuscript, "accident" was modified to "incident." |

Round 2
Reviewer 1 Report
Much better.
Reviewer 3 Report
The authors have addressed all suggestions